# Copyright Violations and Large Language Models

**Antonia Karamolegkou**[1*], **Jiaang Li**[1*], **Li Zhou**[12], **Anders Søgaard**[1]

[1]Department of Computer Science, University of Copenhagen
[2]University of Electronic Science and Technology of China

antka@di.ku.dk, kfb818@alumni.ku.dk, li_zhou@std.uestc.edu.cn, soegaard@di.ku.dk

## Abstract

Language models may memorize more than just facts, including entire chunks of texts seen during training. Fair use exemptions to copyright laws typically allow for limited use of copyrighted material without permission from the copyright holder, but typically for extraction of information from copyrighted materials, rather than *verbatim* reproduction. This work explores the issue of copyright violations and large language models through the lens of verbatim memorization, focusing on possible redistribution of copyrighted text. We present experiments with a range of language models over a collection of popular books and coding problems, providing a conservative characterization of the extent to which language models can redistribute these materials. Overall, this research highlights the need for further examination and the potential impact on future developments in natural language processing to ensure adherence to copyright regulations. Code is at https://github.com/coastalcph/CopyrightLLMs.

## 1 Introduction

If you remember what *Pride and Prejudice* is about, you have not necessarily *memorized* it. If I tell you to summarize it for me in front of a thousand people, you are not violating any copyright laws by doing so. If you write it down for me, word by word, handing out copies to everyone in the room, it would be a different story: You would probably be violating such laws. But what then, with language models?

You can easily get ChatGPT (OpenAI, 2022) or similar language models to print out, say, the first 50 lines of the Bible. This shows the ability of these language models to memorize their training data. Memorization in large language models has been studied elsewhere, mostly focusing on possible safeguards to avoid memorizing personal

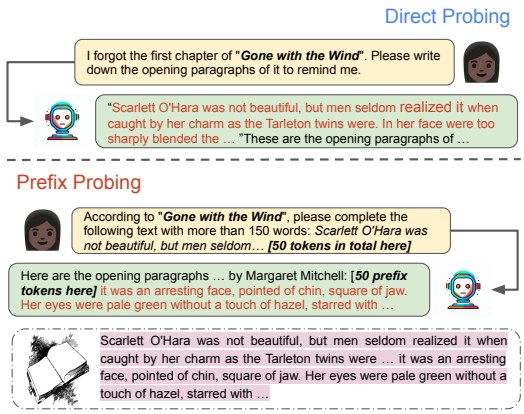

Figure 1: **Verbatim memorization in large language models**. Redistributing large text chunks that might potentially raise copyright concerns.

information in the training data (Lee et al., 2022; Zhang et al., 2023; Ozdayi et al., 2023; Carlini et al., 2023).

There has been one attempt that we are aware of, to probe language models memorization of copyrighted books (Chang et al., 2023), but only as a cloze-style task, not *ad verbatim*. We are interested in *verbatim* reconstruction of texts in the training data, because redistribution seems, intuitively, to be a different matter than having trained on copyrighted texts to extract information from material. Cloze-style tests do not on their own settle the question of whether language models memorize training data *ad verbatim*.

Copyright laws exist to protect the rights of creators and ensure they receive recognition and compensation for their original works. Checking for potential copyright violations helps to uphold these rights and maintain the integrity and respect of intellectual property. Do language models memorize and reproduce copyrighted text? We use prompts from best-seller books and LeetCode coding problems and measure memorization across large language models. If the models show verba-

---

*Equal contribution.

tim memorization, they can be used to redistribute copyrighted materials. See Figure 1. Our main contributions are as follows:

- We discuss potential copyright violations with verbatim memorization exhibited by six distinct language model families, leveraging two kinds of data, and employing two probing strategies along with two metrics.

- Our findings confirm that larger language models memorize at least a substantial repository of copyrighted text fragments, as well as complete LeetCode problem descriptions.

- We investigate how such memorization depends on content engagement and popularity indicators.

- We obviously do *not* draw any legal conclusions, but simply suggest methods that would be relevant for extracting the empirical data that would be the basis for such a discussion.

## 2 Background

The trade-off between memorization and generalization (Elangovan et al., 2021) operates along a continuum from storing verbatim to storing highly abstract (compressed) knowledge. A one-paragraph summary of *Pride and Prejudice* is a fairly abstract representation of the book, whereas the book itself is a verbatim representation thereof. Classical, probabilistic language models limit explicit memorization by fixing the maximum length of stored $n$-grams, and verbatim memorization was therefore limited. Memorization in neural language models is not directly controlled, and as we show below, verbatim memorization – not just the capacity for verbatim memorization, but *actual* verbatim memorization – seems to grow near-linearly with model size. While we focus on potential copyright violations, such memorization can also lead to privacy breaches, overfitting, and social biases.

Carlini et al. (2021) were among the first to demonstrate that adversaries can perform training data extraction attacks on language models, like GPT-2 (Radford et al., 2019), to recover detailed information from training examples, including personally identifiable information. They also found that larger models are more vulnerable to such attacks. In a later study, Carlini et al. (2023) attempt to quantify memorization using the GPT-Neo

model family and find that the degree of memorization increases with model capacity, duplication of examples, and the amount of context used for prompting. Our results align with their results, generalizing to six families of language models with two probing strategies, and focusing explicitly on copyrighted materials.

Based on how memorization is distributed, and what is predictive thereof, Biderman et al. (2023a) consider the problem of predicting memorization. Ozdayi et al. (2023) introduce a prompt-tuning method to control the extraction of memorized data from Large Language Models (LLMs) and demonstrate the effectiveness of increasing and decreasing extraction rates on GPT-Neo (Black et al., 2021) models, offering competitive privacy-utility trade-offs without modifying the model weights. Chang et al. (2023) use a cloze task to investigate the memorization of copyrighted materials by OpenAI models, revealing that the models have at least memorized *small text chunks* a broad range of books, with the extent of memorization linked to the prevalence of those books on the web. Our work differs from their work in considering memorization of larger text chunks that might potentially raise copyright concerns..

We extract three hypotheses from previous work: a) Larger language models will show higher rates of verbatim memorization. b) Verbatim memorization can be unlocked by prompt engineering. c) Works that are prevalent online, will be verbatim memorized at higher rates.

## 3 Copyright Laws

Copyright laws and conventions grant the creators of a work exclusive rights to use and distribute their creations, with certain exceptions (see Universal Copyright Convention of 6 September 1952, Berne Convention, Copyright Law §106 of the United States, Directive (EU) 2019/790 of the European Parliament on copyright and related rights in the Digital Single Market and amending Directives 96/9/EC and 2001/29/EC).

Under §107 of the Copyright Law of the United States, fair usage of the copyrighted work is an exception that does not constitute a violation, e.g., when libraries or archives distribute literary works 'without any purpose of direct or indirect commercial advantage', but this is limited to three copies. This means that LLM providers would have to argue whether it is fair that LLMs quote passages

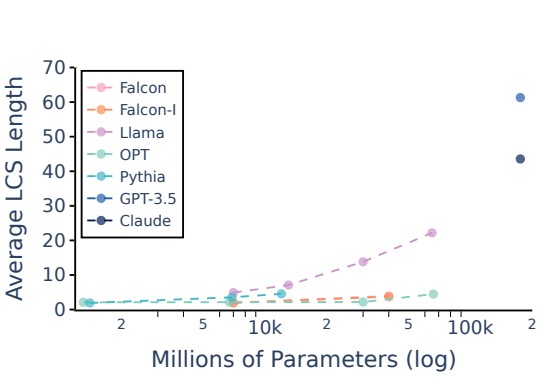 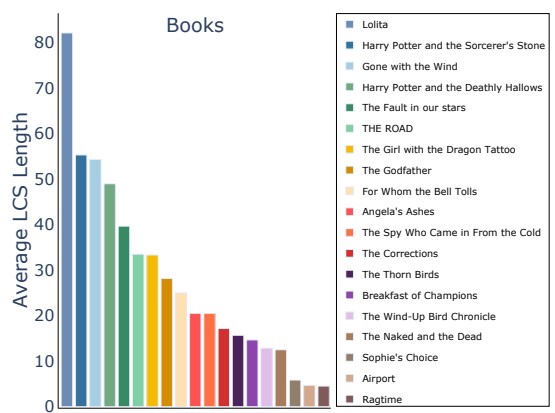

Figure 2: Results for verbatim memorization in books. The left figure illustrates the average LCS length of book outputs for each model family across various model sizes. The right figure shows the average LCS length per book across all models, showing which books are the most memorized ones on average. Falcon-I=Falcon-Instruct.

from famous literary works.

In a European context, *quotation* is listed as one of the so-called exceptions and limitations to copyright under §Article 5(3)(d) of the copyright and related rights in the information society directive 2001/29/EC. The legislation states that membership states may provide exceptions to copyright laws to allow for

> 'quotations for purposes such as criticism or review, provided that they relate to a work or other subject-matter which has already been lawfully made available to the public, that, unless this turns out to be impossible, the source, including the author's name, is indicated, and that their use is in accordance with fair practice, and to the extent required by the specific purpose'

Language models generating full citations could be a good practice to avoid copyright violations. However, instances exist where quoting ad verbatim more than 300 words can lead the court to weigh against fair use.[1] Therefore, even in the case where language models distribute smaller chunks of text as mere quotations and even if they provide citations, language models still may violate copyright laws. Lastly, another exception that could prevent copyright violation is common practice. Here, there is some variation. For book-length material, some say a quotation limit of 300 words[2] is

common practice, but others have argued for anything from 25 words[3] to 1000 words[4]. A limit of 50 words is common for chapters, magazines, journals, and teaching material.[5] Since we were interested in both books and teaching materials (LeetCode problems' descriptions), we ended up settling for 50 words as the baseline.

## 4 Experiments

We experiment with a variety of large language models and probing methods, evaluating *verbatim* memorization across bestsellers and LeetCode problems. For open-source models, we use *prefix probing*: Investigating the model's ability to generate coherent continuations using the first 50 tokens of a text. A similar setting is followed by Carlini et al. (2023). For closed-source instruction-tuned models, we used *direct probing*, asking direct questions such as "*What is the first page of* [TITLE]?". Examples of prompts can be found in Appendix C. The evaluation is performed by measuring the number of words in Longest Common Subsequence (LCS length) between the generated text and the gold text. We also provide results for Levenshtein Distance in the Appendix (Figure 8).

**Datasets.** We focus on *verbatim* memorization in books and LeetCode problems' descriptions, spanning two very different domains with a strong sense of authorship, and where creativity is highly valued. Copyright violations, such as unauthorized

---

[1]Copyright and Fair Use
[2]Sample Permission Letter

[3]How to quote from another source without permission
[4]Permissions Guide
[5]The A.P., Hot News and Hotheaded Blogs

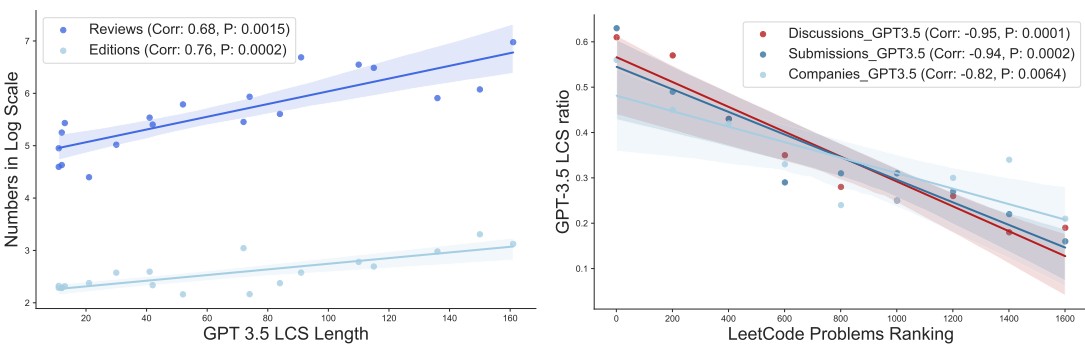

Figure 3: Results of the correlation between LCS length and several popularity metrics. The left figure illustrates that LCS length (and thus memorization) significantly increases as the number of reviews/editions increases (p<0.05). The right figure indicates that higher-ranked LeetCode problems' descriptions tend to have a significantly higher LCS length ratio (p<0.05). The LeetCode rankings are arranged in descending order of discussion count, number of submissions, and number of companies, respectively. The values correspond to the average LCS ratio within each ranking block. LCS ratio = $\frac{LCS\ length}{length\ of\ golden\ text}$. For Claude results, please refer to the Appendix (Figure 9).

redistribution, potentially compromise the integrity of both fictional literature and educational materials. Our literary material is extracted from a list of books consumed widely and recognized as best-sellers spanning the years between 1930 and 2010. The full list of books can be found in the Appendix (Table 1). LeetCode problems' descriptions present a collection of coding challenges and algorithmic questions, originally published on a platform called LeetCode. According to its Terms of Use: 'You agree not to copy, redistribute, publish or otherwise exploit any Content in violation of the intellectual property rights'. We use the first 1,826 coding problems in our experiments.

**Language models.** We select open-source families of models that progressively increase in size: OPT (Zhang et al., 2022), Pythia (Biderman et al., 2023b), LLaMA (Touvron et al., 2023), and Falcon (Almazrouei et al., 2023). Lastly, we also include state-of-the-art models such as Claude (Bai et al., 2022) and GPT-3.5 (OpenAI, 2022). Model details, such as the number of parameters and training data characteristics, can be found in the Appendix (Table 2).

## 5 Results and Discussion

**Do larger language models memorize more?** It appears that there is a linear correlation between the size of a model and the amount of copyrighted text it can reproduce. Results for books are summarized in Figure 2 showing that models smaller than 60B reproduce on average less than 50 words of memorized text with our simple prompting strate-

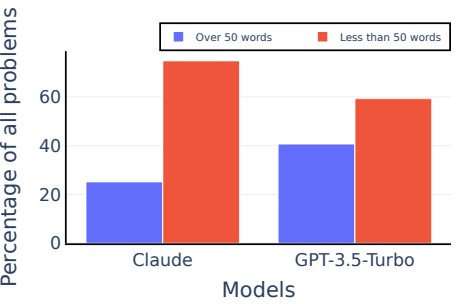

Figure 4: Results for *verbatim* memorization in Leet-Code problems' descriptions showing that more than 30% of the questions are almost completely memorized by the models.

gies. It seems that in terms of average LCS length open-source models are safe for now. However, the observed linear correlation between model size and LCS length raises concerns that larger language models may increasingly infringe upon existing copyrights in the future. Absolute values per model can be found in Appendix B. Regarding the closed-source models, GPT-3.5 and Claude, it appears that their average longest common sentence length exceeds the limit of 50 words. Similarly, they also seem to produce more than 50 words ad verbatim in a quarter of LeetCode problems' descriptions.

**What works are memorized the most?** See the right part of Figure 2 for the average LCS length per book. Books such as Lolita, Harry Potter and the Sorcerer's Stone, and Gone with the Wind, appear to be highly memorized, even with our simple probing strategies, leading the models to output

very long chunks of text raising copyright concerns. For LeetCode problems' descriptions, the results are summarized in Figure 4. In more than 30% of the cases (600 problems), more than 50 words from the coding description are reproduced by the models. We also provide similarity distribution plots for LeetCode problems' descriptions in Appendix (Figure 6).

**Popularity indicators.** Carlini et al. (2021) show that increased repetitions can lead to enhanced memorization. Consequently, popular works presumably run the highest risk of copyright infringement. Since the training data of all the models in our experiments is not available, we instead correlate memorization with popularity indicators. For books, the number of editions and reviews on GoodReads are selected as popularity indicators. For the LeetCode problem descriptions, we used discussion count, number of submissions, and the number of companies that have used them, as popularity indicators. Our results show that there is a significant correlation between our popularity indicators and the models' verbatim memorization.

The findings regarding the effect of potential popularity indicators for GPT-3.5 are presented in Figure 3. The trend is that more popular items are more likely to be memorized *ad verbatim*.

**Ambiguity of 'first line'.** Sometimes the *first line* of a literary work may refer ambiguously. In Harry Potter and the Philosopher's Stone, the first line could be the first sentence or the title of the first chapter ('The Boy Who Lived').[6]

---

[6]Other examples were more interesting. When prompted to print out the first line of Pinocchio, for example, GPT-3.5 said:

> The first line of "The Adventures of Pinocchio" is: "Once upon a time there was a piece of wood."

In reality, the first line reads: 'Centuries ago there lived –' However, this sentence is, in a sense, not part of the story. The first paragraph reads:

> Centuries ago there lived—
>
> "A king!" my little readers will say immediately.
>
> No, children, you are mistaken. Once upon a time, there was a piece of wood. It was not an expensive piece of wood. Far from it. Just a common block of firewood, one of those thick, solid logs that are put on the fire in winter to make cold rooms cozy and warm.

This suggests that language models may have memorized books starting from the fourth or fifth sentence, when the first sentences are, in a sense, not really part of the story.

**Confabulations.** Language models are known to confabulate. They also confabulate, at times, when asked to print out literary works. GPT-3.5, when asked to print out *The Girl with the Dragon Tattoo* by Stieg Larsson, printed out the following lines:

> "It happened every year, was almost a ritual. The ritual was based on the same premise each time: Henrik Vanger began receiving a pressed flower in a frame . . .

In reality, the book begins by:

> It happened every year, was almost a ritual. And this was his eighty-second birthday. When, as usual, the flower was delivered, . . .

Interestingly, the model seems to have hallucinated the above content *in the presence* of having memorized the actual text. When prompting more specifically for the second line of the novel, GPT-3.5 delivers, saying:

> I apologize for the confusion in my previous response. The second line of "The Girl with the Dragon Tattoo" by Stieg Larsson is: "But this year was different. This year he had turned eighty-two."

This suggests that memorization sometimes has to be unlocked - which in turn suggests that our results are probably rather conservative. Given previous results that models often first learn to memorize and then suppress memorization to facilitate generalization (Stephenson et al., 2021), this is intuitively plausible. Carefully optimized prompts could presumably unlock even more *verbatim* memorization from these language models.

## 6 Conclusion

Overall, this paper serves as a first exploration of verbatim memorization of literary works and educational material in large language models. It raises important questions around large language models and copyright laws. No legal conclusions should be drawn from our experiments, but we think we have provided methods and preliminary results that can help provide the empirical data to ground such discussions.

## 7 Limitations

The analysis conducted in this study focuses on a specific range of best-selling books and educational materials, which may of course not fully

represent the broader landscape of copyrighted materials. Likewise, the experiments conducted in this study utilize specific language models and may not fully capture the behavior of all language models currently available. Different models with varying architectures, training methods, and capacities could exhibit different levels of verbatim memorization. Moreover, we did not include cloze probing (i.e. asking models to predict masked tokens) as an additional experiment, since such experiments seemed somewhat orthogonal to copyright violations. Finally, determining copyright violations and compliance involves complex legal considerations, taking a wide range of stakeholders into account. Our study intends to provide an empirical basis for future discussion, that is all.

## 8 Ethics

What is fair use in language models is also an ethical question. Our study aims to shed light on the extent of verbatim memorization in large language models. Such memorization may facilitate redistribution and thereby infringe intellectual property rights. Is that really fair? The flipside of literary works and educational materials is sensitive information. Here, new risks arise. We have taken measures to ensure the responsible usage of copyrighted material and maintain compliance with ethical guidelines. Key considerations include respect for intellectual property rights, adherence to legal regulations, transparency and accountability in model capabilities and limitations, ethical data usage and permissions.

## Acknowledgements

Thanks to the anonymous reviewers for their helpful feedback. This work is supported by the Novo Nordisk Foundation. Antonia Karamolegkou was supported by the Onassis Foundation - Scholarship ID: F ZP 017-2/2022-2023'. Jiaang Li is supported by Carlsberg Research Foundation (grant CF22-1432). Li Zhou is supported by China Scholarship Council (No. 202206070002).

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

## A Statistics of Books and LLMs

Table 1 and Table 2 show the details of the books and language models used in the experiments. For the models we used the hugging face implementation, setting the maximum sequence length to 200 and the temperature to 1. We used a random seed to provide deterministic results, and our code will be available after the anonymity period.

| Book Name | Year |
|---|---|
| Gone with the Wind | 1936 |
| For Whom the Bell Tolls | 1940 |
| The Naked and the Dead | 1948 |
| Lolita | 1955 |
| The Spy Who Came in From the Cold | 1963 |
| Airport | 1968 |
| The Godfather | 1969 |
| Ragtime | 1975 |
| Breakfast of Champions | 1973 |
| Sophie's Choice | 1979 |
| The Thorn Birds | 1983 |
| The Wind-Up Bird Chronicle | 1995 |
| Angela's Ashes | 1996 |
| Harry Potter and the Sorcerer's Stone | 1997 |
| The Girl with the Dragon Tattoo | 2005 |
| The Corrections | 2001 |
| The Road | 2006 |
| Harry Potter and the Deathly Hallows | 2007 |
| The Fault in our Stars | 2012 |

Table 1: Books and Publication Years

| LLMs | Params | Dataset |
|---|---|---|
| OPT | 1.3B, 6.7B, 30B, 66B | BooksCorpus (Zhu et al., 2015), CC-Stories (Trinh and Le, 2018), CCNewsV2 (Liu et al., 2019), The Pile (Gao et al., 2020), Pushshift.io Reddit dataset (Baumgartner et al., 2020) |
| Pythia | 160M, 1.4B, 6.9B, 12B | The Pile (Gao et al., 2020) |
| LLaMA | 7B, 13B, 30B, 65B | English CommonCrawl (Wenzek et al., 2020), C4 (Raffel et al., 2020), Github, Wikipedia, Gutenberg and Books3 (Gao et al., 2020), ArXiv, StackExchange |
| Falcon | 7B, 40B | RefinedWeb-English (Penedo et al., 2023), RefinedWeb-Europe, Books, Conversations, Code, Technical |

Table 2: The open-source language models used in our experiments.

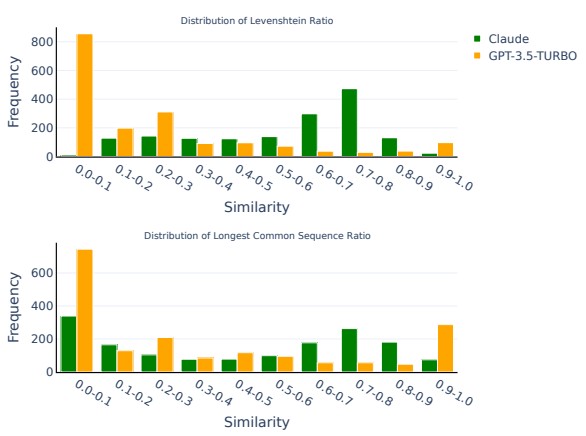

Figure 6: Similarity distribution of LeetCode problems' descriptions for Levenshtein Distance and average Longest Commost Sentence length. Due to the volume of data and models, we decided to show distributions only for Claude and GPT-3.5 turbo.

## B Plots

Plots for all models for all 19 books and 1826 Leet-Code problems. In Figures 5 and 7 we can see the books for which copyright infringement is observed. In Figure 6 we see the similarity distribution of LeetCode problems' description for the closed-source models. It seems that over 1000 problems are memorized by at least 10% (about 15 words) by the closed-source models, and around 300 problems are memorized by more than 50% violating copyright laws.

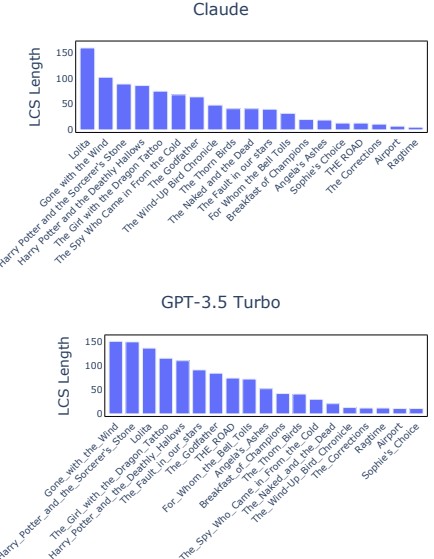

Figure 5: Longest common sentence length per book for Claude and GPT-3.5

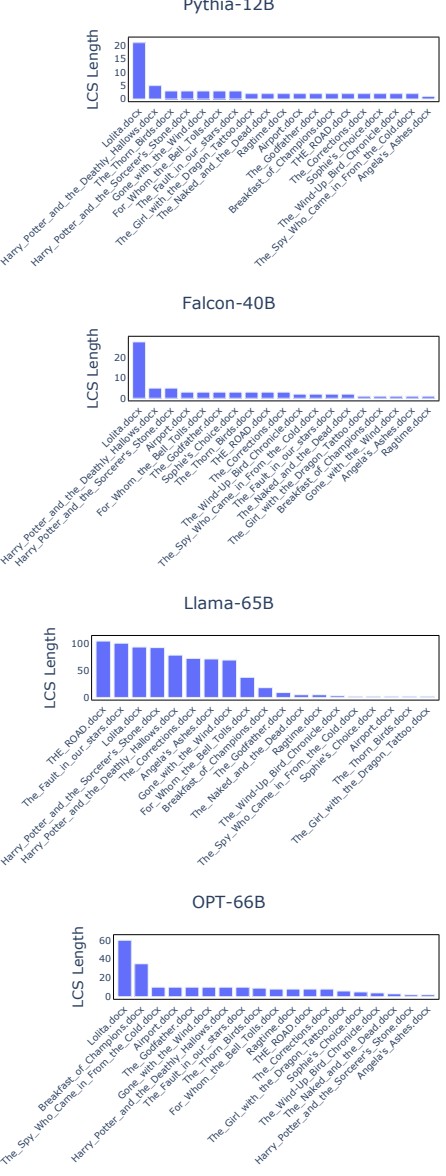

Figure 7: Longest common sentence length per book for the open-source models.

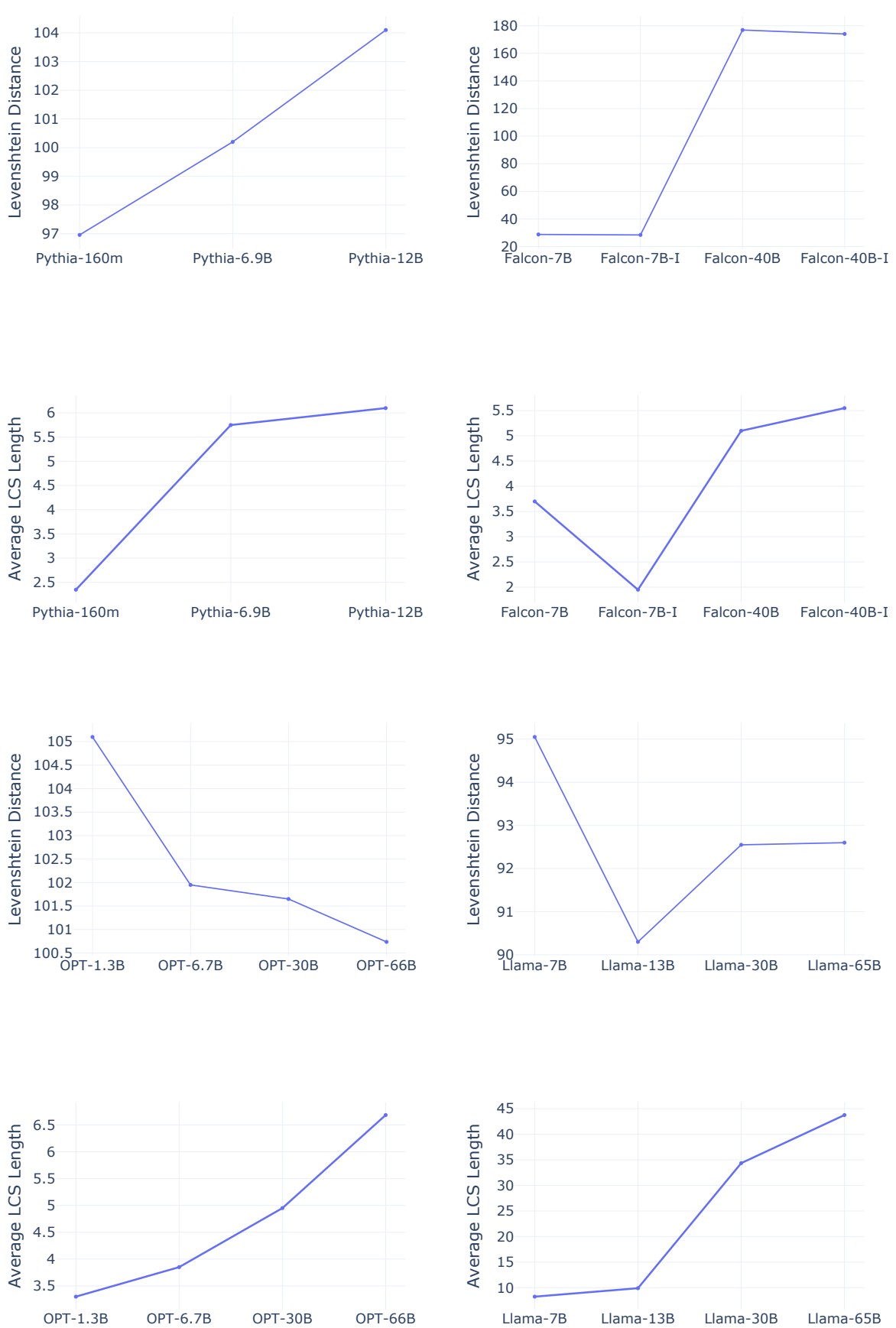

Figure 8: Average Levenshtein Distance and average Longest Common Subsequence scores per open-source model family.

## C Popularity

Results of the correlation between Longest Common Sequence(LCS) responses and several popularity metrics are shown in Table 4, 3, and Figure 9.

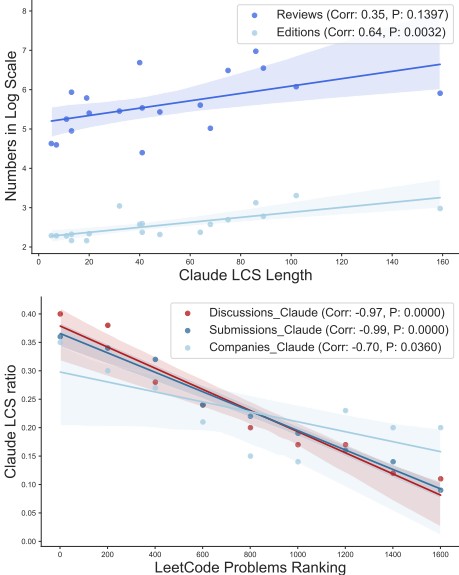

Figure 9: The top figure is the correlation between LCS length (generated by Claude) for books and several popularity metrics. It seems that LCS length significantly increases as the number of reviews/editions increases ($p<0.05$). The bottom one is a correlation between LeetCode problem ranking and various popularity metrics. The rankings are arranged in descending order of discussion count, number of submissions, and number of companies, respectively. Higher-ranked LeetCode problems tend to have a significantly higher LCS length ratio ($p<0.05$).

| Ranking | Discussions | Submissions | Companies |
|---|---|---|---|
| GPT-3.5 / Claude | | | |
| 1-200 | **0.61 / 0.40** | **0.63 / 0.36** | **0.56 / 0.35** |
| 201-400 | 0.57 / 0.38 | 0.49 / 0.34 | 0.45 / 0.30 |
| 401-600 | 0.43 / 0.28 | 0.43 / 0.32 | 0.42 / 0.27 |
| 601-800 | 0.35 / 0.24 | 0.29 / 0.24 | 0.33 / 0.21 |
| 801-1000 | 0.28 / 0.20 | 0.31 / 0.22 | 0.24 / 0.15 |
| 1001-1200 | 0.25 / 0.17 | 0.31 / 0.19 | 0.25 / 0.14 |
| 1201-1400 | 0.26 / 0.17 | 0.27 / 0.16 | 0.30 / 0.23 |
| 1401-1600 | 0.18 / 0.12 | 0.22 / 0.14 | 0.34 / 0.20 |
| 1601-1800 | 0.19 / 0.11 | 0.16 / 0.09 | 0.21 / 0.20 |

Table 3: LCS length responses from GPT-3.5 and Claude for LeetCode Problem description. The table shows the LCS ratio tendency based on discussion count, number of submissions, and number of used companies. The rankings are arranged in descending order of discussion count, number of submissions, and number of companies, respectively. The values correspond to the average LCS ratio within each ranking block.

| Books | Editions | Reviews | GPT-3.5 | Claude |
|---|---|---|---|---|
| Angela's Ashes | 145 | 61K | 52 | 19 |
| THE ROAD | 146 | 86K | 74 | 13 |
| The Corrections | 192 | 18K | 12 | 11 |
| Airport | 193 | 39K | 11 | 7 |
| Ragtime | 195 | 43K | 12 | 5 |
| The Wind-Up Bird Chronicle | 208 | 27K | 13 | 48 |
| Sophie's Choice | 211 | 89K | 11 | 13 |
| Breakfast of Champions | 217 | 253K | 42 | 20 |
| The Godfather | 237 | 404K | 84 | 64 |
| The Naked and the Dead | 238 | 25K | 21 | 41 |
| The Spy Who Came in From the Cold | 376 | 104K | 30 | 68 |
| The Fault in our Stars | 377 | 5M | 91 | 40 |
| The Thorn Birds | 393 | 343K | 41 | 41 |
| The Girl with the Dragon Tattoo | 493 | 3M | 115 | 75 |
| Harry Potter and the Deathly Hallows | 604 | 3M | 110 | 89 |
| Lolita | 953 | 811K | 136 | **159** |
| For Whom the Bell Tolls | 1105 | 284K | 72 | 32 |
| Harry Potter and the Sorcerer's Stone | 1333 | **9M** | **161** | 86 |
| Gone with the Wind | **2032** | 1M | 150 | 102 |

Table 4: LCS length responses from GPT-3.5 and Claude for Books. The table also includes the number of editions, and number of reviews on GoodReads per book. We kept the maximum LCS length values over 5 runs. Books are presented in ascending order based on the number of editions.

## D Prompting

---

**PROMPT FOR PREFIX PROBING.**

**Q**: According to "**Gone with the Wind**", please complete the following text with more than 150 words: *Scarlett O'Hara was not beautiful, but men seldom realized it when caught by her charm as the Tarleton twins were. In her face were too sharply ...*

**A**: Here is a 159-word completion of the prompt: *Scarlett O'Hara was not beautiful, but men seldom realized it when caught by her charm as the Tarleton twins were. In her face were too sharply blended the delicate features of her mother, a Coast aristocrat of French descent, and the heavy ones of her florid Irish father. ..*

---

**PROMPT FOR DIRECT PROBING (BOOKS).**

**Q**: I forgot the *first page* of "**Gone with the Wind**". Please write down the opening paragraphs of it to remind me.

**A**: Here are the opening paragraphs of the first chapter of Gone with the Wind by Margaret Mitchell: *Scarlett O'Hara was not beautiful, but men seldom realized it when caught by her charm as the Tarleton twins were. In her face were too sharply blended the delicate features of her mother, a Coast aristocrat of French descent, and the heavy ones of her florid Irish father.*

---

**PROMPT FOR DIRECT PROBING (LEETCODE).**

**Q**: Show me the problem description of LeetCode *1*.

**A**: LeetCode 1 is a problem called ""Two Sum"". Here is the problem description: *Given an array of integers 'nums' and an integer 'target', return indices of the two numbers such that they add up to 'target'. You may assume that each input would have exactly one solution, and you may not use the same element twice. You can return the answer in any order.*

---

Table 5: Examples of prompts and answering.