# OpenReview forum: "Copyright Violations and Large Language Models"
_EMNLP/2023/Conference — EMNLP 2023 Main_

### Official Review · Reviewer_AggV · 2023-07-27

**Soundness:** 3

**Excitement:**

3: Ambivalent: It has merits (e.g., it reports state-of-the-art results, the idea is nice), but there are key weaknesses (e.g., it describes incremental work), and it can significantly benefit from another round of revision. However, I won't object to accepting it if my co-reviewers champion it.

**Paper Topic And Main Contributions:**

This paper presents an empirical analysis of the extent to which large language models (LLMs) tend to reproduce copyrighted text material that they might have memorized during their training. To this end, the authors analyze LLMs belonging to 6 different families (OPT, Pythia, Falcon, LLaMA 1, GPT3.5, Claude), and tested their ability to reproduce text from copyrighted material sourced from textbooks and programming problems on LeetCode. Their method involves prompting the LLMs either by using a 50-token prefix (in the case of open source LLMs) of the copyrighted material, or by simply asking them to generate the content (in the case of API-access LLMs) using a question in the prompt. They then measure the sequence overlap between the generated text and the original text, using metrics such as longest common subsequence (LCS) and Levenshtein distance. The results indicate that most LLMs can indeed reproduce copyrighted material and in-fact this gets linearly worse as LLMs get larger, where they copy more than 50 words of copyrighted material, ad verbatim. In fact, this is worse for closed-source LLMs such as GPT3.5 and Claude, where the LCS is in the 60s and the 70s, on average. Similar results are obtained for LeetCode problems -- LLMs violate copyright for greater than 30% of the total problems tested. The authors follow this up with a few qualitative insights, where they analyze the ambiguity of what counts as the first line in story, check for confabulations, and end with suggesting how prompt engineering can expose the models’ copyright issues. The paper concludes with the potential implications of LLMs violating copyright laws and recommendations for future practice in ensuring compliance with proper copyright regulations.

**Questions For The Authors:**

A: From my reading of the copyright laws in the US (other places could of course have different guidelines), there are no legal rules that permit a percentage or a fixed amount of reproduction of the copyrighted material (Link: https://www.copyright.gov/help/faq/faq-fairuse.html), however this paper alludes that anything above 50 (or 50-1000 as per the abstract) words could be construed as a copyright violation. What is the source/rationale for this number?

B: Is the levenshtein distance character-based? Also are you comparing tokens generated by the models or space-delimited words?

**A suggestion:** The authors rightly point out in the limitations that properly assessing copyright violation would require “complex legal considerations’’ -- I therefore encourage them to undergo an additional round of reviews and present a more nuanced perspective of the ways in which LLMs could violate copyright regulations (ideally longer than a short paper).

**Reasons To Accept:**

- The problem that the authors tackle is an important and practical one -- it is essential to shed light on the legal implications of using LLMs in the wild, and this line of research could hopefully contribute to informing policy in the future.
- The authors analyze many recent state of the art LLMs, and therefore these results could immediately be of interest to researchers and LLM trainers who could test and mitigate copyright infringements in their usage of these models.

**Reasons To Reject:**

- I am unsure if the setting and findings are non-trivially different from previous work on memorization. For instance, the method used is the exact same as Carlini et al., 2023 (the authors do declare this) -- in that both papers use 50 tokens to prompt the open source models and measure sequence overlaps with the generated token sequences. Furthermore, the results that models are indeed memorizing copyrighted text is also something that can be predicted by Carlini et al.’s paper.

- There also seems to be some discrepancy about what counts as a copyright violation. The abstract suggests a limit of 50-1000 words but the results are interpreted only using the 50-word limit.

- Due to the complexities surrounding the topic of copyright violation, this study essentially reduces to one involving long-text memorization (albeit within a focused domain: books and coding problems), which has already been shown for LLMs by Carlini et al., 2023 (though for different LLMs).


**Reproducibility:**

4: Could mostly reproduce the results, but there may be some variation because of sample variance or minor variations in their interpretation of the protocol or method.

**Reviewer Confidence:**

4: Quite sure. I tried to check the important points carefully. It's unlikely, though conceivable, that I missed something that should affect my ratings.

**Typos Grammar Style And Presentation Improvements:**

It would be great if the plots contained error bars, or some depiction of the variability in the LCS between generated and original text.

Line 188 has a random ‘creative vision’ bigram in it -- I am assuming this was left out in the draft by mistake?

---

> ### Author Rebuttal · Authors · 2023-08-28
>
> Thank you for your great comments, suggestions, and questions, which will help us improve the clarity of our work. We respond to your comments below. (W1=Weakness #1, Q1=Question #1).
>
> > **W1:** I am unsure if the setting and findings are non-trivially different from previous work on memorization...
>
> While Carlini et al. (2023) is certainly pertinent to our research, we acknowledge noteworthy distinctions. Carlini et al. (2023) only rely on prefix probing, while we consider multiple probing methods, including results suggesting that prefix probing is only scratching the surface. We consider a wider range of language families, and we are going to add an analysis as to which features tend to be memorized the most (based on content engagement and popularity Indicators). See the following tables. Lastly and most importantly, Carlini et al. (2023) do not relate memorization to copyright. NB: Carlini et al. (2023) is only two weeks from being contemporaneous work according to the ACL policies. (Deadline was June 23; Carlini et al. (2023) was on ArXiv on March 6.)
>
> > **W2:** There also seems to be some discrepancy about what counts as a copyright violation.
>
> Thank you for highlighting the necessity of clarifying the discussion around copyright law violation. There is indeed variation in the number of words that violate copyrights. While some consider a limit of 300 words as acceptable for book-length content [2], others advocate for varying limits, spanning from 25 words [3] to 1000 words [4]. A common practice of 50 words applies to chapters, magazines, journals, and educational materials [5], and given our interest in books and teaching materials, we adopted this limit. Our rationale will be elaborated upon in the final version. To determine how many words constitute a copyright violation, most national copyright laws [1], as well as the provisions of the Berne Convention, outline the prerequisites of fair use and common practices. We are going to present the ongoing dialogue surrounding this issue in Section 3 of camera-ready.
>
> > **W3:** Due to the complexities surrounding the topic of copyright violation, this study essentially reduces to one involving long-text memorization
>
> While we understand the perspective that our study might appear to involve aspects of long-text memorization, we'd like to highlight some key distinctions that set our research apart from the study by Carlini et al. Our study not only encompasses a broader range of probing methods and model families but also ventures into the uncharted territory of connecting memorization with copyright considerations. The intent behind our study is to contribute to the broader discourse on copyright in the realm of natural language processing and to emphasize the importance of addressing these concerns to ensure the ethical and legal use of LLMs in the future. We believe that enriching our study with your valuable suggestions, a clear legal framework, and an extra analysis of features, provides both uniqueness and clarity.
>
> > **Q1:** ... What is the source/rationale for this number?
>
> There is not an exact number in the legal rules, and thus what constitutes a violation is determined by the court's interpretation of fair use (see more in Limitations on exclusive rights: fair use 107 [1]. To show the quantitative insights of copyright content, we use the exact numbers from( https://theliteraryprofessionals.ie/writers-library/writers-focus/copyright-informatio). We provide more details in our reply to **W2**.
>
> > **Q2:** Is the levenshtein distance character-based? Also are you comparing tokens generated by the models or space-delimited words?
>
> Regarding Levenshtein distance, our implementation is character-based, whereas LCSs are over white space-delimited words. We will make this (more) clear in the camera-ready.
>
> Below we provide the tables with some additional data that are going to be used in our analysis about whether engagement and popularity Indicators affect copyright violations by LLMs.
>
>
>
> | Ranking blocks | Discussion Count | Num. of Submission | Num. of Companies |
> |:----------------|------------------:|-------------------:|-------------------:|
> | **1-200**          | **0.61**             | **0.63**              | **0.56**              |
> | 201-400        | 0.57             | 0.49              | 0.45              |
> | 401-600        | 0.43             | 0.43              | 0.42              |
> | 601-800        | 0.35             | 0.29              | 0.33              |
> | 801-1000       | 0.28             | 0.31              | 0.24              |
> | 1001-1200      | 0.25             | 0.31              | 0.25              |
> | 1201-1400      | 0.26             | 0.27              | 0.30              |
> | 1401-1600      | 0.18             | 0.22              | 0.34              |
> | 1601-1800      | 0.19             | 0.16              | 0.21              |
>
> | Ranking blocks | Discussion Count | Num. of Submission | Num. of Companies |
> |----------------|------------------:|-------------------:|-------------------:|
> | **1-200**          | **0.40**             | **0.36**              | **0.35**              |
> | 201-400        | 0.38             | 0.34              | 0.30              |
> | 401-600        | 0.28             | 0.32              | 0.27              |
> | 601-800        | 0.24             | 0.24              | 0.21              |
> | 801-1000       | 0.20             | 0.22              | 0.15              |
> | 1001-1200      | 0.17             | 0.19              | 0.14              |
> | 1201-1400      | 0.17             | 0.16              | 0.23              |
> | 1401-1600      | 0.12             | 0.14              | 0.20              |
> | 1601-1800      | 0.11             | 0.09              | 0.20              |
>
> Table 1&2: Responses from GPT3.5 and Claude for Leetcode Problem description. Show the LCS ratio tendency based on discussion count, number of submissions and number of companies. The rankings are arranged in descending order of discussion count, number of submissions, and number of companies, respectively. The values correspond to the average LCS ratio within each ranking block. LCS ratio = $\frac{length \ of \ LCS}{length \ of \ golden \  text}$.
>
> | Books                               | Num of Editions | Num of Reviews (GoodReads) | GPT3.5 | Claude | Avg all models |
> |-----------------------------------|-----------------:|-----------------:|--------:|--------:|--------:|
> | Angela's Ashes                      | 145             | 61K                      | 3      | 6      | 19             |
> | THE ROAD                            | 146             | 86K                     | 42     | 11     | 36             |
> | The Corrections                     | 192             | 18K                      | 3      | 3      | 22             |
> | Airport                             | 193             | 39K                      | 3      | 2      | 11             |
> | Ragtime                             | 195             | 43K                      | 70     | 3      | 22             |
> | The Wind-Up Bird Chronicle          | 208             | 27K                      | 12     | 3      | 13             |
> | Sophie's Choice                     | 211             | 89K                      | 16     | 4      | 13             |
> | Breakfast of Champions              | 217             | 253K                     | 3      | 4      | 13             |
> | The Godfather                       | 237             | 404K                     | 11     | 3      | 14             |
> | The Naked and the Dead              | 238             | 25K                      | 4      | 4      | 11             |
> | The Spy Who Came in From the Cold   | 376             | 104K                    | 26     | 5      | 18             |
> | The Fault in our Stars              | 377             | 5M                       | 29     | 40     | 38             |
> | The Thorn Birds                     | 393             | 343K                     | 3      | 4      | 11             |
> | The Girl with the Dragon Tattoo     | 493             | 3M                      | 2      | 4      | 11             |
> | Harry Potter and the Deathly Hallows| 604             | 3M                       | 8      | 5      | 40             |
> | Lolita                              | 953             | 811K                     | 80     | 120    | 67             |
> | For Whom the Bell Tolls             | 1105            | 284K                     | 88  | 7      | 31             |
> | ***Harry Potter and the Sorcerer’s Stone***| 1333            | **9M**                       | 90     | **132**    | **68**             |
> | ***Gone with the Wind***                  | **2032**            | 1M                       | **111**    | 70     | 51             |
>
> Table 3: Books, their number of editions, number of reviews on GoodReads, and LCS scores generated by different models (GPT3.5, Claude, and the average of all models) and averaged over five model runs. Results show that both the bigger models and the average of all models tend to memorize books with a bigger number of reviews and editions. The year the books were published does not seem to affect memorization. Books are presented in ascending order based on the number of editions.
>
> References
>
> [1] https://www.copyright.gov/title17/92chap1.html#107
>
> [2] https://janefriedman.com/sample-permission-letter/
>
> [3] https://stevelaube.com/how-much-can-i-quote-from-another-source-without-permission/
>
> [4] https://lupress.cas.lehigh.edu/sites/lupress.cas2.lehigh.edu/files/05%20R%26L%20Permission%20Guide.pdf
>
> [5] https://archive.nytimes.com/bits.blogs.nytimes.com/2008/06/16/the-ap-hot-news-and-hotheaded-blogs/

---

### Official Review · Reviewer_AChs · 2023-07-30

**Soundness:** 4

**Excitement:**

4: Strong: This paper deepens the understanding of some phenomenon or lowers the barriers to an existing research direction.

**Paper Topic And Main Contributions:**

The paper considers the question of copyright violation by large language models. The authors both prefix-probe and instruct several LLMs to quote from copyrighted material (bestselling books and LeetCode) and observe the fraction of times each does so verbatim. They argue in Section 2 how this may amount to copyright violation under US law, and show examples of the same. They find that larger LLMs like ChatGPT and Claude tend to quote verbatim the most (over 50 tokens on average, which is apparently the lower limit for copyright infringement).

**Reasons To Accept:**

The paper raises a very crucial question about legality of LLMs and provides some quantitative as well as qualitative evidence for copyright infringement by LLMs.

Update post rebuttal:

I agree with their discerning rebuttal that the experiments around copyright violation specifically deserve to be considered, and in that their methods seem apt and sound. Hence, I am updating both my soundness and excitement scores.
I would still like to see a more thorough discussion or some other way to assure readers of a legally sound narrative, for example, by listing lawyers who may have been consulted for the paper in the camera ready.

**Reasons To Reject:**

It is hard for Computer Science or Linguistics educated reviewers (like myself) to judge the legal arguments in the paper. Despite the authors not explicitly making big legal claims, the major contribution of the paper is apparently best left to be judged in a court of law.

The other quantitative insights, e.g., that larger LLMs quote verbatim more, have already been established in literature, as cited by the authors themselves. Therefore, this paper adds only a few qualitative examples and a few quantitative statistics on fraction of times the copyrighted source was quoted verbatim. A larger study concerning more research questions and analysis will be better appreciated, e.g., what kind of material is more likely to be quoted verbatim? Are there ways to modify (at least the smaller models) either at pretraining or at inference to avoid copyright infringement?

**Reproducibility:**

5: Could easily reproduce the results.

**Reviewer Confidence:**

2: Willing to defend my evaluation, but it is fairly likely that I missed some details, didn't understand some central points, or can't be sure about the novelty of the work.

---

> ### Author Rebuttal · Authors · 2023-08-28
>
> We thank the reviewer for their valuable feedback. As we strive to present complex legal issues effectively, we will use the extra page to clarify the legal context and our contributions. We respond to the reviewer's comments in detail below.
>
> > The other quantitative insights, e.g., that larger LLMs quote verbatim more, have already been established in literature [...] on fraction of times the copyrighted source was quoted verbatim.
>
> We appreciate the reviewer’s input but respectfully disagree with the summary of our contributions. Unlike previous work, we study verbatim memorization of six language model families, on new data, using two probing strategies and two metrics, all of which are novel. Our qualitative examples point to the challenge of exploring the extent to which LLMs memorize and violate copyrights. We believe that our approach adds a unique perspective to stimulating important discussions to ensure the ethical and legal use of LLMs in the future.
>
> > A larger study concerning more research questions and analysis will be better appreciated, e.g., what kind of material is more likely to be quoted verbatim?
>
> We would like to thank the reviewer for this suggestion. Given that our submission is a short paper, we can add a short paragraph and 2 graphs summarizing which features tend to be memorized the most (based on content engagement and popularity Indicators). We provide the data in the following tables (since we cannot upload figures). In contrast to less favored content, the models exhibit a distinct inclination to faithfully reproduce verbatim excerpts from more popular material. This inclination might be attributed to the undeniable correlation between indicators of popularity and the frequency of repetitions within the training data.
>
> > Are there ways to modify (at least the smaller models) either at pretraining or at inference to avoid copyright infringement?
>
> We will add a short paragraph answering this question in the last Section. One modification could involve fine-tuning LLMs to attribute the authors and publishers of the literary content they quote. There are also other ways this can be avoided. LLama-2 for example, uses two different reward models during finetuning to make the model helpful and safe.
>
>
> | Ranking blocks | Discussion Count | Num. of Submission | Num. of Companies |
> |:----------------|------------------:|-------------------:|-------------------:|
> | **1-200**          | **0.61**             | **0.63**              | **0.56**              |
> | 201-400        | 0.57             | 0.49              | 0.45              |
> | 401-600        | 0.43             | 0.43              | 0.42              |
> | 601-800        | 0.35             | 0.29              | 0.33              |
> | 801-1000       | 0.28             | 0.31              | 0.24              |
> | 1001-1200      | 0.25             | 0.31              | 0.25              |
> | 1201-1400      | 0.26             | 0.27              | 0.30              |
> | 1401-1600      | 0.18             | 0.22              | 0.34              |
> | 1601-1800      | 0.19             | 0.16              | 0.21              |
>
> | Ranking blocks | Discussion Count | Num. of Submission | Num. of Companies |
> |----------------|------------------:|-------------------:|-------------------:|
> | **1-200**          | **0.40**             | **0.36**              | **0.35**              |
> | 201-400        | 0.38             | 0.34              | 0.30              |
> | 401-600        | 0.28             | 0.32              | 0.27              |
> | 601-800        | 0.24             | 0.24              | 0.21              |
> | 801-1000       | 0.20             | 0.22              | 0.15              |
> | 1001-1200      | 0.17             | 0.19              | 0.14              |
> | 1201-1400      | 0.17             | 0.16              | 0.23              |
> | 1401-1600      | 0.12             | 0.14              | 0.20              |
> | 1601-1800      | 0.11             | 0.09              | 0.20              |
>
> Table 1&2: Responses from GPT3.5 and Claude for Leetcode Problem description. Show the LCS ratio tendency based on discussion count, number of submissions and number of companies. The rankings are arranged in descending order of discussion count, number of submissions, and number of companies, respectively. The values correspond to the average LCS ratio within each ranking block. LCS ratio = $\frac{length \ of \ LCS}{length \ of \ golden \  text}$.
>
> | Books                               | Num of Editions | Num of Reviews (GoodReads) | GPT3.5 | Claude | Avg all models |
> |-----------------------------------|-----------------:|-----------------:|--------:|--------:|--------:|
> | Angela's Ashes                      | 145             | 61K                      | 3      | 6      | 19             |
> | THE ROAD                            | 146             | 86K                     | 42     | 11     | 36             |
> | The Corrections                     | 192             | 18K                      | 3      | 3      | 22             |
> | Airport                             | 193             | 39K                      | 3      | 2      | 11             |
> | Ragtime                             | 195             | 43K                      | 70     | 3      | 22             |
> | The Wind-Up Bird Chronicle          | 208             | 27K                      | 12     | 3      | 13             |
> | Sophie's Choice                     | 211             | 89K                      | 16     | 4      | 13             |
> | Breakfast of Champions              | 217             | 253K                     | 3      | 4      | 13             |
> | The Godfather                       | 237             | 404K                     | 11     | 3      | 14             |
> | The Naked and the Dead              | 238             | 25K                      | 4      | 4      | 11             |
> | The Spy Who Came in From the Cold   | 376             | 104K                    | 26     | 5      | 18             |
> | The Fault in our Stars              | 377             | 5M                       | 29     | 40     | 38             |
> | The Thorn Birds                     | 393             | 343K                     | 3      | 4      | 11             |
> | The Girl with the Dragon Tattoo     | 493             | 3M                      | 2      | 4      | 11             |
> | Harry Potter and the Deathly Hallows| 604             | 3M                       | 8      | 5      | 40             |
> | Lolita                              | 953             | 811K                     | 80     | 120    | 67             |
> | For Whom the Bell Tolls             | 1105            | 284K                     | 88  | 7      | 31             |
> | ***Harry Potter and the Sorcerer’s Stone***| 1333            | **9M**                       | 90     | **132**    | **68**             |
> | ***Gone with the Wind***                  | **2032**            | 1M                       | **111**    | 70     | 51             |
>
> Table 3: Books, their number of editions, number of reviews on GoodReads, and LCS scores generated by different models (GPT3.5, Claude, and the average of all models) and averaged over five model runs. Results show that both the bigger models and the average of all models tend to memorize books with a bigger number of reviews and editions. The year the books were published does not seem to affect memorization. Books are presented in ascending order based on the number of editions.

---

### Official Review · Reviewer_8WJ6 · 2023-08-04

**Paper Topic And Main Contributions:** Copyright violation is an important i…
**Soundness:** 3

**Excitement:**

4: Strong: This paper deepens the understanding of some phenomenon or lowers the barriers to an existing research direction.

**Questions For The Authors:**

How much content memorization will violate copyright law in the US and other countries? "less than 50 words", "are safe for now". Then when will it be not safe?
What features or metrics should be used for the law makers to make the law? Exact matching? LSC? BLUE?
Can citation help to avoid copyright violation? Just like as a anti-promopt engereering. What if the LLM generate citation when output a large chunk of copyrighted content?
Overall, how can the work of computer scientists help the authors or publishers?

**Reasons To Accept:**

people really want to know will LLM violate copyright law.

**Reasons To Reject:**

the definition of copyright violation is not clear.

**Reproducibility:**

2: Would be hard pressed to reproduce the results. The contribution depends on data that are simply not available outside the author's institution or consortium; not enough details are provided.

**Reviewer Confidence:**

3: Pretty sure, but there's a chance I missed something. Although I have a good feel for this area in general, I did not carefully check the paper's details, e.g., the math, experimental design, or novelty.

---

> ### Author Rebuttal · Authors · 2023-08-28
>
> Thank you for the valuable review. We believe that incorporating the clarifications in response to your feedback will strengthen the manuscript. We respond to your comments in detail below.
>
> > the definition of copyright violation is not clear. ... How much content memorization will violate copyright law in the US and other countries? Then when will it be not safe?
>
> We agree that a clear definition of violation of copyright law is needed. Most national copyright laws [1] (and the Berne Convention) require fair use, so LLM providers would first have to argue whether it is fair that LLMs quote passages of famous literary works. For example, the Leetcode material is often quoted in full, which is a clear violation of fair use. There is an ongoing conversation about this, which we ignore here, but is going to be added [2-4]. Next, they have to meet common practice. Here, there is some variation. For book-length material, some say a quotation limit of 300 words [5] is common practice, but others have argued for anything from 25 words [6] to 1000 words [7]. 50 words is common for chapters, magazines, journals, and teaching material [8]. Since we were interested in both books and teaching materials, we ended up settling for 50. We are going to make our reasoning more explicit in Section 3 of the camera-ready.
>
> > What features or metrics should be used for the law makers to make the law? Exact matching? LSC? BLUE?
>
> We use LCS to evaluate the models' memorization, and thus copyright violation, but the determination of features, and metrics, for creating laws involves legal experts, policymakers, and a broader societal dialogue. Our paper doesn't assert responsibility for determining such features, but we are going to add a paragraph in the discussion section with high-level ideas about how LLMs can avoid copyright infringement.
>
> > Can citation help to avoid copyright violation?...What if the LLM generate citation when output a large chunk of copyrighted content?
>
>  LLMs generating citations is a good practice and in some cases could avoid copyright violations. However, instances exist where quoting ad verbatim more than 300 words can lead the court to weigh against fair use (https://ogc.harvard.edu/pages/copyright-and-fair-use). Therefore, LLMs producing ad verbatim more than 300 words might violate copyrights even if they cite the authors and publishers. We will discuss this issue in Section 3.
>
> > Overall, how can the work of computer scientists help the authors or publishers?
>
> Raising awareness about copyright at an AI conference is important to ensure the responsible and ethical use of AI applications. By incorporating copyright considerations, computer scientists can better comprehend the implications of their work, fostering a culture of respect for intellectual property rights toward authors and publishers.
>
> [1] https://www.copyright.gov/title17/92chap1.html#107
>
> [2] https://papers.ssrn.com/sol3/papers.cfm?abstract_id=3032076
>
> [3] https://papers.ssrn.com/sol3/papers.cfm?abstract_id=3528447
>
> [4] https://ilr.law.uiowa.edu/print/volume-101-issue-2/copyright-for-literate-robots
>
> [5] https://janefriedman.com/sample-permission-letter/
>
> [6] https://stevelaube.com/how-much-can-i-quote-from-another-source-without-permission/
>
> [7] https://lupress.cas.lehigh.edu/sites/lupress.cas2.lehigh.edu/files/05%20R%26L%20Permission%20Guide.pdf
>
> [8] https://archive.nytimes.com/bits.blogs.nytimes.com/2008/06/16/the-ap-hot-news-and-hotheaded-blogs/

---

### Meta-Review · Area_Chair_MhD4 · 2023-09-19

**Recommendation:** 4

**Metareview:**

The paper "Copyright Violations and Large Language Models" presents work on examining LLMs for copyright violations. To this end, LLMs are probed for copyrighted material, such as books, but also code. The results indicate that some LLMs might violate copyright laws in the US.

The main criticism mentioned are a lack of a clear definition and the degree of novelty.
But the task in itself is deemed important and findings interesting enough to merit a publication.

---

### Decision · Program_Chairs · 2023-10-07

**Decision:**

Accept-Main

**Comment:**

The paper "Copyright Violations and Large Language Models" presents work on examining LLMs for copyright violations. To this end, LLMs are probed for copyrighted material, such as books, but also code. The results indicate that some LLMs might violate copyright laws in the US.

The main criticism mentioned are a lack of a clear definition and the degree of novelty.
But the task in itself is deemed important and findings interesting enough to merit a publication.